# Wearable Inertial Sensor Approach for Postural Adjustment Assessments during Predictable Perturbations in Sport

**DOI:** 10.3390/s22218272

**Published:** 2022-10-28

**Authors:** Manuela Brito Duarte, Anderson Antunes da Costa Moraes, Eduardo Veloso Ferreira, Gizele Cristina da Silva Almeida, André dos Santos Cabral, Anselmo de Athayde Costa e Silva, Daniela Rosa Garcez, Givago da Silva Souza, Bianca Callegari

**Affiliations:** 1Laboratório de Estudos da Motricidade Humana, Av. Generalíssimo Deodoro 01, Belém 66073-00, PA, Brazil; 2Centro de Ciências Biológicas e da Saúde, Universidade do Estado do Pará, Tv. Perebebuí, 2623—Marco, Belém 66087-662, PA, Brazil; 3Programa de Pós Graduação em Ciências do Movimento, Universidade Federal do Pará, Av. Generalíssimo Deodoro 01, Belém 66073-00, PA, Brazil; 4University Hospital Bettina Ferro de Souza, Federal University of Pará, R. Augusto Corrêa, n1, Belém 66075-110, PA, Brazil; 5Instituto de Ciências Biológicas, Universidade Federal do Pará, Rua Augusto Corrêa 01, Belém 66075-110, PA, Brazil; 6Núcleo de Medicina Tropical, Universidade Federal do Pará, Avenida Generalíssimo Deodoro 92, Belém 66055-240, PA, Brazil

**Keywords:** postural control, wearables, sports

## Abstract

Introduction: Evidence supports the importance of efficient postural control to improve performance in sports. This involves the use of strategies such as anticipatory posture adjustments and compensatory adjustments. Technology makes analysis and assessments in sports cheaper, while being valid and reliable compared to the gold-standard assessment equipment. Objectives: This article aimed to test the validity and reliability of signals extracted from the sensor’s accelerometer (*Metamotion*
*C*), by comparing it to the data obtained from the gold-standard equipment (a three-dimensional video-motion-capture system). Design: Observational, cross-sectional study. Methods: We exposed 20 healthy young standing people to the pendulum impact paradigm, which consisted of predictable anteroposterior disturbances applied at the shoulder level. In order to measure this, we observed the acceleration of the center of mass in the anticipatory and compensatory phase of the disturbance and compared the signals of the two devices (*Metamotion*
*C* and a motion-capture system). Results: The validation results showed the significant linear correlation of all variables with a moderate to large correlation of r ≥ 0.5 between the devices. In contrast, the reliability results between sessions obtained by filming were all significant and above 0.75, indicating excellent reliability. The APAonset variable had a reasonable to high intra-class correlation in the anticipatory phase. In the compensatory phase, the CPA_time_ variable showed an excellent correlation. Conclusions: *Metamotion*
*C* proved reasonably valid and highly reliable in measuring the center of mass acceleration compared to the camera system in both the anticipatory and compensatory phases.

## 1. Introduction

Performance in sport demands efficient postural control and the use of strategies such as anticipatory posture adjustments (APAs) and compensatory adjustments (CPAs) [1]. APAs are characterized by the displacement of the center of pressure (COP) and the advanced activation or inhibition of postural muscles in the face of a predictable disturbance. After the disturbance occurs, the compensatory strategy is used, which regulates the COP back to the stability position; this event is guaranteed, subserved by the feedback mechanism that acts on the restoration of balance and muscle activation [2]. These concepts have already been studied in sport. Evidence supports the importance of anticipatory programming to improve reaction and response stability in swimming and soccer athletes and also contact sports such as rugby, among others [3,4].

Postural adjustment assessments have been restricted, for years, to research laboratories, fully equipped with force platforms, surface electromyography, and motion-capture systems. These are expensive and unwieldy solutions. In this context, the use of low-cost portable technologies, as an alternative, has been employed in the sports environment [5]. Wearable sensors, such as inertial sensors (IMU), may enable the assessment of movement patterns during activities in real-life sport settings and based on the desire for the constant monitoring and quantification of results and developments, these sensors have been rising in popularity [6].

Although some studies have already demonstrated the validity and reliability of inertial sensors to access sports performance [7] and during functional activity [8,9], the current limitations of IMU validation and reliability research are still present. A recent review of 82 papers demonstrated the excellent validity and reliability of IMUs for mean spatiotemporal parameters during walking, but they call for caution in using joint angle measurement and other biomechanical outcomes such as stability, regularity, and segmental accelerations [10].

*Metamotion C* (mBientLab, *Metamotion* C, San Francisco, CA, USA) is a commercial triaxial IMU and, together with other similar IMUs from the same manufacturer, have been used in studies in sports, although in a restricted (i.e., no testing validity) and scarce way [11]. Recently, excellent accuracy in this device was found in classifying martial arts movements, having placed it on the torso of volunteers and evaluated the movements performed during the fight [12,13]. In tennis athletes, this IMU was attached to their wrist to investigate precision results related to ball speed and rotation. The results showed that the device could accurately recognize a player’s action concerning the ball [14]. Finally, placing this IMU inside a ball and three other sensors attached to the upper back and wrist of bowlers in another study evaluated the correlation of video-motion-capture marker positions with acceleration data. These results highlighted the device’s ability to record the biomechanical characteristics of the bowler to complement the analysis in the game of bowling [15].

The use of IMUs to assess APAs can enable low-cost assessments outside of research laboratories in the sports field, which is still less explored. The present study is aimed to test the validity and reliability of a commercially used *Metamotion C* accelerometer (mBientLab, *Metamotion* C, San Francisco, CA, USA) to extract accelerometer signals and quantify APAs and CPAs during a predictable perturbation. We aim to validate the signals extracted from the sensor’s accelerometer and compare them with the data obtained from gold-standard equipment (a three-dimensional camera system). The hypothesis is that the data extracted from the *Metamotion C* sensor accelerometer can evaluate the APAs and CPAs equally as well as the gold-standard method and show the reliability of measurements between different sessions.

## 2. Methods

### 2.1. Study Design and Ethical Considerations

In this cross-sectional observational study, we investigated the concurrent validity and reliability of an IMU wearable device and a video-motion-capture system. The procedures performed were approved by the local Research Ethics Committee (CEP No. 3,817,332), as well as the Observational Studies in Epidemiology (STROBE) Statement. All participants were informed about the investigation procedures and signed a consent form to participate in the study.

### 2.2. Subjects

For this study, we recruited 20 healthy young people (10 men and 10 women), with a mean height of 1.68 ± 0.081 m, a mean age of 29.57 ± 6.66 years, and a mean weight of 72.79 ± 14.32 kg. Participants with a history of orthopedic, neurological, or rheumatic problems or any other disease that could interfere with task performance were excluded from the study. Participants were recruited on demand. The sample size was calculated using 80% statistical power and a 95% confidence interval. The mean and standard deviation for the APAonset (s) was estimated in a pilot study performed on the first 7 subjects in two sessions. The mean difference between sessions was 0.012 ± 0.011 s. A required sample size of 10 individuals was calculated, and the authors decided to test the double of this minimum.

### 2.3. Postural Adjustments Evaluation

Participants were placed barefoot and shoulder-width apart from each other with *Metamotion C* and a reflective marker positioned in the fifth lumbar vertebra region (L5) [16,17]. Initially, the subjects were instructed to jump vertically in place. The alignment of the recordings to the signal peak on the vertical axis, which characterizes the moment of impact with the ground, was used to synchronize the signals of the two assessment instruments. After this step, the subjects were positioned again, in front of a pendulum fixed to the ceiling, containing a second reflective marker for the video-motion-capture recording of the pendulum’s movement. The pendulum consisted of a height-adjustable central shaft with the distal ends protected by two padded pieces positioned shoulder-width apart. A load (3% of the participant’s body weight) was attached to the distal end of the central rod, above the padded pieces (Figure 1).

The experimental protocol consisted of an anteroposterior perturbation, caused by the unidirectional force applied to the torso of the participants by the impact of the pendulum, which was released by the researcher from a distance of 0.5 m from the subject. Participants were able to observe the pendulum during the experiment, ensuring the predictability of disturbance. Twelve trials were performed, with random time intervals between them for each participant (0.5–4 s), to avoid any “training effect” during the trials. After two weeks, the same experiment was conducted, with the same volunteers, to analyze the reproducibility of the data.

To record the COM accelerations, two instruments were used: a three-dimensional video-motion-capture system with three cameras (Simi Motion, Simi, Unterschleißheim, Germany), with a sampling frequency of 120 Hz, and a Metamotion C wireless inertial sensor (mBientLab, Metamotion C, San Francisco, CA, USA). Metamotion C is a sensor that records raw sensor data via Bluetooth up to 400 Hz and transmits this data up to 100 Hz and has an approximate weight of 28.34 g and dimensions height × width × depth (in mm): 8 × 25.4 × 27. This equipment has a light sensor, a temperature sensor, and a sensor fusioned with 10 axes of motion detection (3 axes accelerometer + 3 axes gyroscope + 3 axes magnetometer + altimeter/barometer/pressure). In this study, data were collected from the device’s accelerometer, along the three X, Y, and Z axes, at a frequency of 100 Hz and exported in CSV format through the Metabase application (mBientLab, San Francisco, CA, USA) provided by the inertial sensor developers.

### 2.4. Data Analysis and Variables

Data synchronization, processing, and analysis were performed offline using MATLAB R2020 software (MathWorks, Natick, MA, USA). The impact of the pendulum on the subject’s torso was defined as the beginning of each disturbance (Time zero, New Castle, DE, USA). The kinematic data (from the video-motion-capture system) of the pendulum trajectory towards the subject’s torso visualized this moment. The time interval was between 200 ms before impact to 400 ms after its analysis. Only COM accelerations in the anteroposterior direction were considered for the outcome measures. The video and accelerometer of the *Metamotion C* sensor were used to generate the raw data coordinates on this axis. They were filtered with a second-order low-pass 30 Hz Butterworth filter, which generated a signal envelope, used for identification by a visual inspection associated with algorithms, of the events to be investigated.

The analyzed variables were the following (see Figure 2):(i)APAonset (APA_onset_): start time of COM acceleration, before Tzero (amplitude greater than the mean of its base value plus 2 standard deviations—SD);(ii)Time to peak acceleration (CPA_time_): time to reach the moment of peak COM acceleration after Tzero.

### 2.5. Statistics

Statistical analysis was performed using GraphPad PRISM 9 and MATLAB (MathWorks, Natwick, MA, USA). The Shapiro–Wilk test confirmed whether the data were distributed normally, and data description was performed using boxplot graphs for each parameter. The boxplot displays the median on the centerline, the top and bottom edges (75th and 25th percentiles), and the endpoints with the minimum and maximum data values. The mean was plotted inside the boxplot.

For the validation of *Metamotion C*, the measured variables were correlated between devices by the Pearson (r) correlation test when they were parametric (CPA_time_) or Spearman (APA_onset_) data. In the correlation tests, the point-to-point agreement between the systems was estimated per subject for each variable of the COM measured, and the estimated r values and confidence intervals were reported. Pearson’s correlation coefficients (r) were interpreted with magnitude thresholds of 0–0.1: trivial; 0.1–0.3: small; 0.3–0.5: moderate; 0.5–0.7: large; 0.7–0.9: very large, and 0.9–1.0: almost perfect [18]. Then, Bland–Altman graphs with 95% confidence limits (mean ± 2 SD) were plotted to compare equipment values.

The reliability between the two sessions and intra session was calculated using an intraclass correlation coefficient (ICC), a standard error of measurement (SEM), and minimal detectable change (MDC) estimates [19]. The intraclass correlation coefficient (ICC) with a mixed model and type of absolute agreement with a 95% confidence interval (CI) was calculated to determine absolute reliability. SEM was calculated using the following formula: $\mathrm{SEM}=\mathrm{SDpooled}\times \sqrt{1-\mathrm{ICC}}$, where SDpooled is the pooled standard deviation. MDC was calculated at the 90% level using the formula ${\mathrm{MDC}}_{90}=\mathrm{SEM}\times \sqrt{2}\times 1.64$. The ICCs were interpreted according to Shrout and Fleiss, wherein ICC ≥ 0.75 indicates excellent correlation, 0.4 ≥ ICC ≥ 0.74 indicates reasonable to high correlation, and ICC ≤ 0.39 indicates poor correlation [20]. The level of significance was defined as *p* < 0.05.

## 3. Results

Figure 3 shows the average of the 12 attempts of each subject and the overall average of the 20 subjects, showing that the studied events can be visualized and characterized using both recording instruments.

The mean of each variable is shown in Figure 4. The boxplots show the median, mean, lower, and upper quartiles and the minimum and maximum values of the anticipatory and compensatory variables, in both sessions, with no differences between the comparisons of equipment and sessions (Figure 4).

The linear correlation of all variables was significant, showing a moderate to large correlation with r ≥ 0.5 between devices (Figure 5). All correlations were statistically significant. Bland–Altman graphs showed similar behaviors between variables with values close to the mean, homogeneous disposition, and reduced dispersion within the limits of agreement.

The ICCs obtained from the filming between sessions were significant and above 0.75, indicating excellent reliability. The APA_onset_ variable had a reasonable to high ICC in the anticipatory phase. In the compensatory phase, the CPA_time_ showed an excellent correlation. APA_onset_ presented SEM and MDC values estimated for video-motion measurements little lower or equivalent to the *Metamotion C*, while CPA_time_ measurements were equivalent (Table 1).

## 4. Discussion and Implication

This study aimed to test the validity and reliability of center-of-mass displacement signals recorded by a commercial accelerometer (*Metamotion C*) by comparing these with signals obtained by a gold-standard system. The hypothesis was based on the premise that the data extracted from *Metamotion C* can evaluate the APAs and CPAs with validation against the gold-standard method and present reliability between sessions. The hypothesis was generally supported, as the linear correlation results presented r ≥ 0.5, demonstrating a moderate to large correlation between the devices and intraclass correlation indices ranging from fair to excellent.

In this study, the pendulum impact paradigm was used to trigger APAs in the participants, given the anticipated perception of the disturbance suffered, with the impact of the load. This experimental task was chosen to reproduce sporting movements (i.e., how a ball is caught during a match) or situations in which APAs may be required to prevent injuries (i.e., in the face of an impact with another athlete or loss of balance). In general, practicing sports involves sudden situations of changes in direction, speed, angle, and dual-task use, which is physiologically challenging and requires the efficient control of COM oscillations, with constant reorganization and postural adaptation [21].

In the present study, the *Metamotion C* sensor was shown to measure the anticipated COM accelerations in a way comparable to the camera system. This brings a positive outlook for its use in sports, as it is portable, low-cost, and independent of laboratory facilities [6]. These are the main reasons for the increasing popularity of IMU use in sports studies. Recently, some papers have demonstrated that the use of portable IMU technology has had a significant impact on athlete monitoring in sports medicine, since it provided physicians, coaches, and training staff with a method of monitoring physiologic and movement parameters during training and competitive sports, in the real-life environments [22].

Although the literature presents a variety of protocols for using IMUs in sports [3,5,23], is important to recognize that this area of research is still under development, especially when we look for evidence supporting the use of wearable sensors to assess CPAs and APAs during a predictable external perturbation in a real sport setting. The difficulty in studying APAs in this environment is one of the reasons why the literature is scarce. Some obstacles to these studies include the difficulty in establishing the exact beginning of a movement or impact, which generates the imbalance, or measuring the displacements of body segments or electrical muscle activity without laboratory equipment for research in human movement (i.e., high-speed video-motion capture, surface electromyography). Thus, studies investigating APAs in sport are more common and present in research laboratory environments. As an exception to this fact, Wang et al. (2018) synchronized wireless electromyography sensors with video-motion-capture recordings during a regular training session of rugby athletes [4]. In this study, it was possible to investigate the anticipatory activities of the neck muscles in real impact situations during training. However, studies such as this have a high cost since they demand technology transfer from the laboratory to the sports environment.

In this context, our results support the possibility of using accelerometers as a low-cost alternative to APA assessments in real-life sport settings. *Metamotion C* is an example of a commercially available sensor; however, as it is a new technology, there is a need to validate its use, and this study contributes to the literature by presenting this data. *Metamotion C,* as a reliable alternative to gold-standard equipment, was supported by the results we found (moderate to large correlation with video-motion measurements and hign to excellent reliability).

Some practical application may derive from these results, since no matter the type of sport, they all highlight the need for athe daptability and reorganization of posture adjustments to maintain balance, lower the risk of injuries, and improve performance. APAs are useful to detect and train position-specific patterns in movement; create more efficient sports-specific training programs for performance optimization; and monitor potential risks of injury, such as concussion and fatigue [3,23].

However, some limitations of the present study must be considered. Generalizing the results to athletes may need further investigation, since the participants in the present study consisted of healthy young people only. Although we have compared signals between two methods of measuring COM acceleration (and no subjects), the literature shows that expertise may affect athletes’ postural strategy, promoting better anticipatory responses [24]. Further investigation including this population would contribute to the generalization. In addition, we used only one (specific) commercial accelerometer model and one environment. Environmental factors (i.e., temperature and friction) may affect the feasibility of the method. The MDC, however, demonstrated that the sensitivity of the *MetamotionC* is comparable to that of the video-motion system and that it would perform well in sports. Further investigation is needed to expand the use of this resource in different devices, groups, environments, and conditions.

## 5. Conclusions

Data recorded by *Metamotion C* were comparable to those of a video-motion-capture system, thus demonstrating that it is a possible tool to be used outside laboratory environments for the evaluation of APAs and CPAs. Moreover, the use of *Metamotion C* as a more environmental and commercially available instrument allows for postural control assessments to be carried out in a more accessible, portable, and cost-effective way.

## Figures and Tables

**Figure 1 sensors-22-08272-f001:**
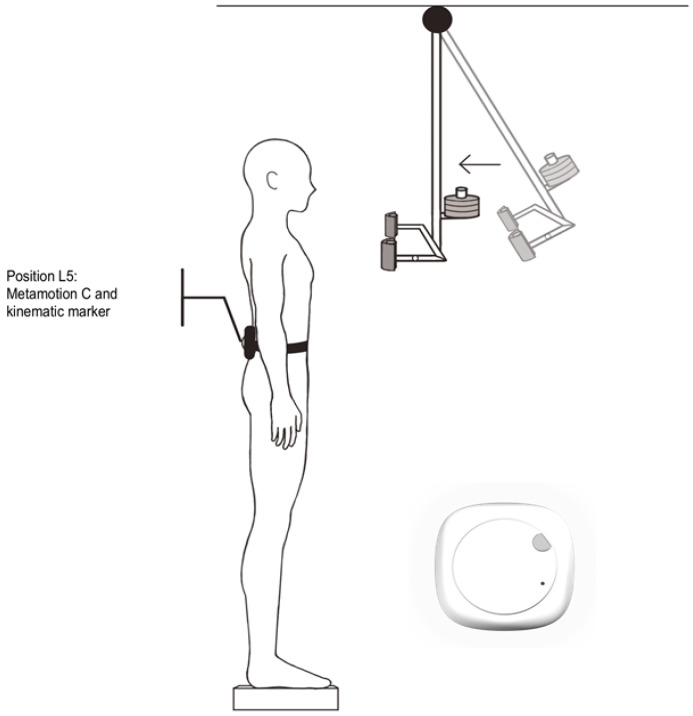
View of the experimental setup showing the participant in front of the pendulum that was released from 0.5 m away, causing a predictable anteroposterior disturbance. In L5, Metamotion C sensor is fixed and the reflective marker on top of it.

**Figure 2 sensors-22-08272-f002:**
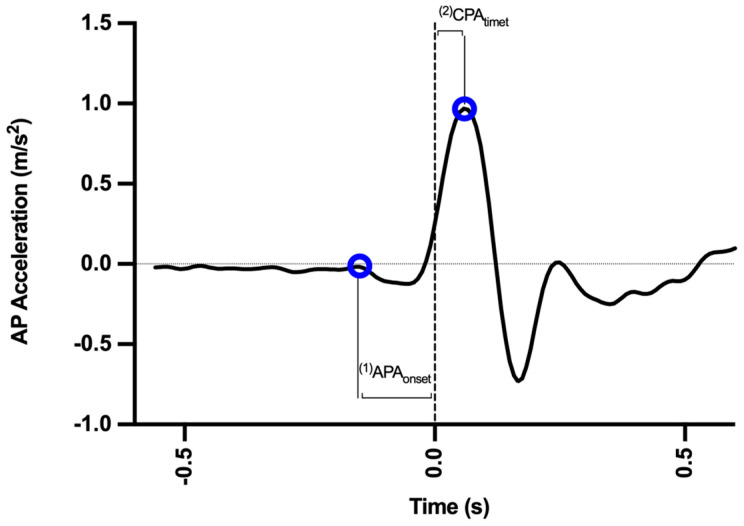
Anteroposterior acceleration curve and measurement variables. The dotted line represents the moment of impact of the pendulum. AP: anteroposterior.

**Figure 3 sensors-22-08272-f003:**
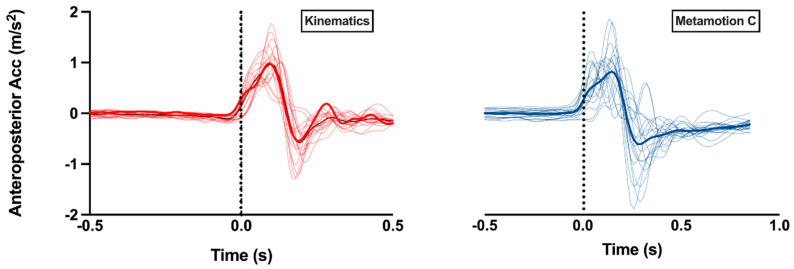
COM anteroposterior acceleration of each subject and the average of all subjects in the first session. Thick red and blue lines represent the average recording resulting from the 20 subjects (in each device. Thin lines represent individual recordings. The dotted line represents the moment of impact of the pendulum. Acc: acceleration.

**Figure 4 sensors-22-08272-f004:**
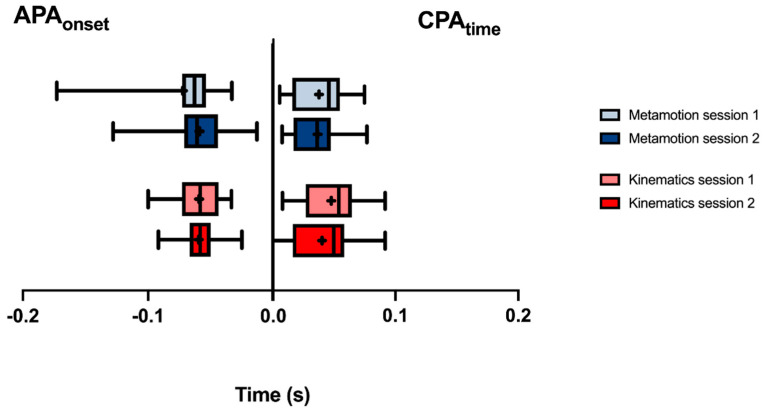
Descriptive analysis of means and standard deviation of anticipatory variables, measured by the video-motion-capture system and *Metamotion C*. Data expressed by the central line = median; box = 25th and 75th percentiles; bars = minimum and maximum values (average values within the box marked X).

**Figure 5 sensors-22-08272-f005:**
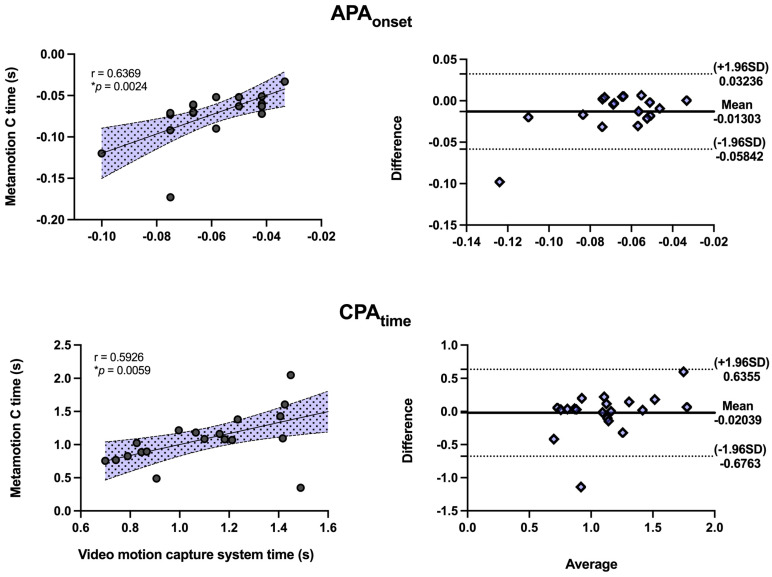
Linear correlation graphs and Bland–Altman correlation graphs show the correlation between evaluation instruments, where r ≥ 0.5 represents a moderate correlation, and the asterisk (*) represents the values that obtained statistical significance (*p* ≤ 0.05).

**Table 1 sensors-22-08272-t001:** The intra-session intraclass correlation coefficient (ICC) of all analyzed variables and the two instruments used, where the asterisk (*) represents the values that obtained statistical significance (*p* ≤ 0.05).

Variable	ICC	Lower Limit	Upper Limit	F	df1	df2	*p*-Value	SEM	MDC
APA_onset_									
Video-motion-capture system	0.745	0.344	0.900	3.788	19	19.02	0.002 *	0.009	0.020
Metamotion C	0.615	0.065	0.845	2.855	19	16.90	0.017 *	0.010	0.041
CPA_time_									
Video-motion-capture system	0.805	0.514	0.922	5.547	19	18.14	0.000 *	0.011	0.026
Metamotion C	0.768	0.406	0.908	4.166	19	19.15	0.001 *	0.010	0.023

## Data Availability

Please contact the corresponding author.

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
