# Peer review of "Wearable Inertial Sensor Approach for Postural Adjustment Assessments during Predictable Perturbations in Sport"

_sensors, 2022, doi:10.3390/s22218272_

Round 1

Reviewer 1 Report

Thank you for the opportunity to review your manuscript, Wearable Inertial Sensor approach for Postural Adjustments assessments during predictable perturbations in sport.

The article's title, before the abstract, must be deleted (lines 21- 22).

Line 23 - Write the objectives in such a way that they coincide with those of the introduction.

A brief introduction in the abstract is necessary to understand the readers better.

Moreover, the general wording is difficult to understand.

Line 62-66. It gives the impression that the aim is to advertise the brand name rather than to describe the existing measurement tools and what this tool provides compared to the existing ones.

Line 67-80. Improve the wording, as it looks like loose notes.

Methodology: The beginning of the methodology should describe the study design and type, not the sample.

No sample size calculation has been carried out.

It goes straight to describing the tools without describing the procedure.

The variables are not explained separately; the measurement tools and the procedure are, but everything is mixed up. Reword it in a way that makes it clear to the readers.

Because the time intervals were randomised for each participant and not equal????. Explain

In the reliability analysis, no data are given for the measurement error and the minimum detectable difference.

Overall the discussion is poor and, at some point, reiterative.

Conclusions are also reiterative and should be rephrased.

Author Response

Dear Editorial Board and reviewers,

The authors have received the evaluation and all considerations were considered. Thank you for helping us to improve the quality of the manuscript. We´ve responded all the comments made, organizing the answer accordingly. Below, there is a list of itemized changes made, addressing each of the revision requirements. All suggestions were attended, and a new version of the manuscript is attached with the changes made (highlighted the within the text in yellow). To improve the language of the manuscript, authors have decided, this time, to have the assistance of a paid editing services.

Reviewer #1:

Dear authors,

Thank you for the opportunity to review your manuscript, Wearable Inertial Sensor approach for Postural Adjustments assessments during predictable perturbations in sport.

  1. The article's title, before the abstract, must be deleted (lines 21- 22).

Authors: Thanks for the contribution. We made changes accordingly

  1. Line 23 - Write the objectives in such a way that they coincide with those of the introduction.

Authors: Thanks for your comment. We have re-phrased the sentence.

  1. A brief introduction in the abstract is necessary to understand the readers better.

Moreover, the general wording is difficult to understand.

Authors: Thanks. We have added a brief background in the abstract and hope it is clearer now.

  1. Line 62-66. It gives the impression that the aim is to advertise the brand name rather than to describe the existing measurement tools and what this tool provides compared to the existing ones.

Line 67-80. Improve the wording, as it looks like loose notes.

Authors: Thanks for helping to improve our background in the manuscript. We have removed the emphasis on the manufacturer/brand and tried to make the rationale behind the use of the sensor proposed, clearer. In addition, the IMU technical information was replace to methods, as suggested by #rev2.

  1. Methodology: The beginning of the methodology should describe the study design and type, not the sample.

Authors: Thanks. We have corrected that.

  1. No sample size calculation has been carried out. 

Authors: Thanks. The sample size was calculated using 80 % statistical power and 95 % confidence interval. The mean and standard deviation for the APAonset (s) was estimated in a pilot study performed in the first 7 subjects in two sessions. The mean difference between sessions was achieved as 0.012 ± 0.011s. A required sample size of 10 individuals was calculated and authors decided to test the double of this minimum. This information was included in “Methods”, in the new version.

  1. It goes straight to describing the tools without describing the procedure.

Authors: Thanks for helping to improve our “Methods” in the manuscript. We have reorganized that.

  1. The variables are not explained separately; the measurement tools and the procedure are, but everything is mixed up. Reword it in a way that makes it clear to the readers.

Authors: Thanks for helping to improve our “Methods” in the manuscript. We have reorganized that, separating subheadings.

  1. Because the time intervals were randomised for each participant and not equal????. Explain

Authors: Thanks for your question. The outcome of previous studies (below) demonstrated that a single training session improves the generation of APAs prior to a predictable external perturbation. So, we decided to randomize the interval between the pendulum releases to avoid this possible “training effect” during the trials. Since all the trials of the pendulum release were implemented by the same experimenter, he changed set the interval between the trials in a range from 0.5 – 4 s, not allowing the same interval between the releases. We tried to explain this better now.

Aruin AS, Kanekar N, Lee YJ, Ganesan M. Enhancement of anticipatory postural adjustments in older adults as a result of a single session of ball throwing exercise. Exp Brain Res. 2015;233:649–655.[PubMed] [Google Scholar]

Kanekar N, Aruin AS. Improvement of anticipatory postural adjustments for balance control: effect of a single training session. J Electromyogr Kinesiol. 2015;25:400–405. [PMC free article] [PubMed] [Google Scholar]

Aruin AS, Kanekar N, Jadghane S. Training-related enhancement of anticipatorty postural adjustments in older adults. In: Hlavacka F, Lobotkova J, editors. 7th International Posture Symposium, Institute of Normal and Pathological Physiology, SAS, Smolenice Castle, Slovak Republic, Europe; 2015. [Google Scholar]

  1. In the reliability analysis, no data are given for the measurement error and the minimum detectable difference.

Authors: Thanks for the contribution. We calculated and provided these data accordingly. Results are in table 2.

  1. Overall the discussion is poor and, at some point, reiterative.

Authors: Thanks. We have rewritten the discussion, providing information on how your results will impact research and/or clinical practice and on the generalizability of the results to the athlete population. We hope it is better now.

  1. Conclusions are also reiterative and should be rephrased.

Authors: Thanks. We have rewritten the conclusion and hope it is better now.

Reviewer 2 Report

The authors present a comparison between the capability and reliability of a Meramotion C sensor unit and a camera based system in detecting the motion of the center of mass of multiple healthy human subjects. The work aims to demonstrate the feasibility of portable inertial sensors for anticipatory posture adjustments (APAs) and compensatory adjustments (CPAs) measurement in sports. That topic can potentially be interesting to the readers, but I feel the scope and depth of the study need to be substantially improved. Specifically:

1. There is no mentioning of the performance specs of the Meramotion C sensors and weather other type of inertial sensors can also be used. If so, what performance level is required for this type of applications.

2. The experiment in this work is very simple and does not represent the complicated scenarios in real sports. The authors emphasize that prior works on motion tracking using inertial sensors are mostly limited to laboratory studies. However, the work presented here hardly provide any additional insights than research works in lab environment by others.

3. The work demonstrates that the accelerometer and reliably detect the motion of the torso, but similar demonstrations have already been widely presented in others works. In this manuscript, there is no quantitative discussion of the performance requirements and limitations of the inertial sensors for the mentioned applications. For example, how would bias drift of the accelerometer affect the measurement accuracy? How does the environmental factors (temperature, friction, impact, etc.) affect the feasibility of the method? For what sports the sensor unit will perform well and for what sports it may see some challenges?

4. How does the mounting location of the sensor affects the detection accuracy of the COM acceleration?

5. Page 3, line 104, it is unclear what are the dimensions of the sensor unit.

6. Figure 3, caption says “(t)hick black lines represent the average recording resulting from the 20 subjects.” There is no thick black line in the Metamotion C plot. And what do the thick red and blue lines represent?

Author Response

Dear Editorial Board and reviewers,

The authors have received the evaluation and all considerations were considered. Thank you for helping us to improve the quality of the manuscript. We´ve responded all the comments made, organizing the answer accordingly. Below, there is a list of itemized changes made, addressing each of the revision requirements. All suggestions were attended, and a new version of the manuscript is attached with the changes made (highlighted the within the text in yellow). To improve the language of the manuscript, authors have decided, this time, to have the assistance of a paid editing services.

Reviewer #3:

Dear authors,

The authors present a comparison between the capability and reliability of a Meramotion C sensor unit and a camera based system in detecting the motion of the center of mass of multiple healthy human subjects. The work aims to demonstrate the feasibility of portable inertial sensors for anticipatory posture adjustments (APAs) and compensatory adjustments (CPAs) measurement in sports. That topic can potentially be interesting to the readers, but I feel the scope and depth of the study need to be substantially improved. Specifically:

  1. There is no mentioning of the performance specs of the Meramotion C sensors and weather other type of inertial sensors can also be used. If so, what performance level is required for this type of applications.

Authors: Thanks for the contribution, this is an excellent point. Inertial sensors such as accelerometers and gyroscopes have already been widely used to monitor exercise performance level. We have included information on this, including a recent systematic review:

Worsey MT, Espinosa HG, Shepherd JB, Thiel DV. Inertial Sensors for Performance Analysis in Sports: A Systematic Review. Sports (Basel). 2019 Jan 21;7(1):28. doi: 10.3390/sports7010028. PMID: 30669590; PMCID: PMC6359075.

However, postural adjustments are difficult to be directly related to performance level, and to try to solve this issue, we chose to calculate measurement error and the minimum detectable difference (SEM and MDC) values, comparing them between the instruments. This information and a rationale around it are included in the new version.

  1. The experiment in this work is very simple and does not represent the complicated scenarios in real sports. The authors emphasize that prior works on motion tracking using inertial sensors are mostly limited to laboratory studies. However, the work presented here hardly provide any additional insights than research works in lab environment by others.

Authors: Thanks for helping to improve our background in the manuscript. We have included new references to address this suggestion. We added studies assessing validity/reliability of accelerations obtained from IMUs during functional/sport activity and also a recent systematic review.

 Cudejko, T.; Button, K.; Amri, M. Al Validity and reliability of accelerations and orientations measured using wearable sensors during functional activities. Sci. Rep. 2022, 1–12, doi:10.1038/s41598-022-18845-x.

Zijlstra, W.; Hof, A.L. Assessment of spatio-temporal gait parameters from trunk accelerations during human walking. Gait Posture 2003, 18, doi:10.1016/S0966-6362(02)00190-X.

Kobsar, D.; Charlton, J.M.; Tse, C.T.F.; Esculier, J.F.; Graffos, A.; Krowchuk, N.M.; Thatcher, D.; Hunt, M.A. Validity and reliability of wearable inertial sensors in healthy adult walking: A systematic review and meta-analysis. J. Neuroeng. Rehabil. 2020, 17.

We hope we have clarified this issue and that the importance of the study, and its adding knowledge is more evident in this new version.

  1. The work demonstrates that the accelerometer and reliably detect the motion of the torso, but similar demonstrations have already been widely presented in others works. In this manuscript, there is no quantitative discussion of the performance requirements and limitations of the inertial sensors for the mentioned applications. For example, how would bias drift of the accelerometer affect the measurement accuracy? How does the environmental factors (temperature, friction, impact, etc.) affect the feasibility of the method? For what sports the sensor unit will perform well and for what sports it may see some challenges?

Authors: Thanks for helping to improve the quality of the manuscript. We tried to address this issue in the new version and hope it is better now. Some of these questions are not yet solved in the literature, and the included systematic reviews present it. We tried to talk about it in the new discussion, when we recommend caution on the generalizability of the results. Temperature, friction must be investigated. However, the MDC calculated demonstrated that the sensitive of the instrument os comparable to kinematics, and would perform well in sports field.

  1. How does the mounting location of the sensor affects the detection accuracy of the COM acceleration? 

 Authors: Thanks for your question. We have addressed this comment by including two new references demonstrating that sensors situated on the low lumbar spine produced greater accuracy than thoracic, or lower limb sensors for static balance analysis. For APAs this comparison was not found in the literature, but we have chosen to use L5 as it is an anticipatory response elapsed to keep the COM within the basis of support, to maintain static balance.

Leiros-Rodriguez R., Arce M.E., Miguez-Alvarez C., Garcia-Soidan J.L. Definitions of the proper placement point for balance assessment with accelerometers in older women. Rev. AndaI. Med. Deporte. 2016 doi: 10.1016/j.ramd.2016.09.001.[CrossRef] [Google Scholar]

Baker N, Gough C, Gordon SJ. Inertial Sensor Reliability and Validity for Static and Dynamic Balance in Healthy Adults: A Systematic Review. Sensors (Basel). 2021 Jul 30;21(15):5167. doi: 10.3390/s21155167. PMID: 34372404; PMCID: PMC8348903.

  1. Page 3, line 104, it is unclear what are the dimensions of the sensor unit.

Authors: Thank, the IMU technical information was added: Dimensions height x width x depth (in mm): 8 x 25.4  x 27.

  1. Figure 3, caption says “(t)hick black lines represent the average recording resulting from the 20 subjects.” There is no thick black line in the Metamotion C plot. And what do the thick red and blue lines represent? 

Authors: Sorry we have corrected this caption. Thick red and blue line represent the average recording resulting from the 20 subjects

Reviewer 3 Report

TITLE

1.     Please add study design, for example “validity and reliability study”.

ABSTRACT

2.     Please remove abbreviations and provide full names

3.     Please also describe reliability part in objectives and methods as currently it is only presented in the results.

4.     Acceleration from the sensors is also kinematics, isn’t it? As such, please replace “kinematics” with “motion capture system” here and throughout the whole paper as this is confusing.

5.     Keywords: please use different; these already occur in the title and abstract.

6.     Practical implications: I would suggest removing this part and discussing this in the discussion. If not, mention only implications of the study to which point 1 does not belong. Generally, in the current form, this is just a repetition of the study results (point 2/3), and “can be employed in sport field” is an overstatement as you assessed the sensor in the controlled laboratory environment,

INTRODUCTION

7.     Lines 58-60: this is confusing, please rephrase

8.     Lines 63-66: this should belong to methods, please replace

9.     In the current form, it is quite difficult to figure out from the information flow in the introduction, why it is important to study this, who will benefit from it, and what is the added value of this paper to current knowledge.

10.  The intro/discussion lacks reference to other similar studies assessing validity/reliability of accelerations obtained from IMUs during functional/sport activities such as https://pubmed.ncbi.nlm.nih.gov/36028523/ and https://pubmed.ncbi.nlm.nih.gov/14654202/ . Please mention/discuss these studies in the intro or discussion.

METHODS

11.  The whole methods section should be rewritten/rearranged according to relevant reporting guidelines, such as STROBE or CONSORT depending on the study design. Information on study design, setting, inclusion and exclusion criteria for study participants, the definition of outcomes, validity, and reliability of instruments, data postprocessing, etc is superficial or lacking and should be provided in separate paragraphs to facilitate reading.

12.  Please justify the sample size in the study.

13.  Line 171: reference 17 is inappropriate here an should be deleted as it only cites reference 13.

DISCUSSION

14.  Lines 247-249: this is just repetition of the results

15.  Please provide information on how your results will impact research and/or clinical practice.

16.  Please discuss the generalizability of the results to the athlete population.

CONCLUSIONS

17.  Line 262: given your study design/population, this is an overestimation – please delete/rephrase.

Author Response

Dear Editorial Board and reviewers,

The authors have received the evaluation and all considerations were considered. Thank you for helping us to improve the quality of the manuscript. We´ve responded all the comments made, organizing the answer accordingly. Below, there is a list of itemized changes made, addressing each of the revision requirements. All suggestions were attended, and a new version of the manuscript is attached with the changes made (highlighted the within the text in yellow). To improve the language of the manuscript, authors have decided, this time, to have the assistance of a paid editing services.

Reviewer #2: To the Authors,

TITLE

  1. Please add study design, for example “validity and reliability study”.

Authors: Thanks for the contribution. We made changes accordingly

ABSTRACT

  1. Please remove abbreviations and provide full names

Authors: Done.

  1. Please also describe reliability part in objectives and methods as currently it is only presented in the results.

Authors: Thanks for your comment. We have re-phrased the sentence, adding this information.

  1. Acceleration from the sensors is also kinematics, isn’t it? As such, please replace “kinematics” with “motion capture system” here and throughout the whole paper as this is confusing.

Authors: Thanks, we have replaced throughout the whole manuscript, including figures and tables.

  1. Keywords: please use different; these already occur in the title and abstract.

Authors: Done.

  1. Practical implications: I would suggest removing this part and discussing this in the discussion. If not, mention only implications of the study to which point 1 does not belong. Generally, in the current form, this is just a repetition of the study results (point 2/3), and “can be employed in sport field” is an overstatement as you assessed the sensor in the controlled laboratory environment,

Authors: Thanks for the contribution. We have removed this section, leaving it to the discussion.

INTRODUCTION

  1. Lines 58-60: this is confusing, please rephrase

Authors: Thanks. We have rephrased lines 58-60 and hope it is better in the new version.

  1. Lines 63-66: this should belong to methods, please replace

Authors: Done.

  1. In the current form, it is quite difficult to figure out from the information flow in the introduction, why it is important to study this, who will benefit from it, and what is the added value of this paper to current knowledge.

Authors: Thanks for helping to improve our background in the manuscript. We have removed the emphasis on the manufacturer/brand and tried to make the rationale behind the use of the sensor proposed, clearer. In addition, the IMU technical information was replace to methods, as suggested.

  1. The intro/discussion lacks reference to other similar studies assessing validity/reliability of accelerations obtained from IMUs during functional/sport activities such as https://pubmed.ncbi.nlm.nih.gov/36028523/

and https://pubmed.ncbi.nlm.nih.gov/14654202/ .  Please mention/discuss these studies in the intro or discussion.

Authors: Thanks for helping to improve our background in the manuscript. We have included these (and a recent systematic review) to clarify this issue. We hope that the importance of the study and its adding knowledge is more evident in this new version.

METHODS

  1. The whole methods section should be rewritten/rearranged according to relevant reporting guidelines, such as STROBE or CONSORT depending on the study design. Information on study design, setting, inclusion and exclusion criteria for study participants, the definition of outcomes, validity, and reliability of instruments, data postprocessing, etc is superficial or lacking and should be provided in separate paragraphs to facilitate reading.

Authors: Thanks for helping to improve our “Methods” in the manuscript. We have reorganized that, addressing the requirements, observing strobe and separating subheadings.

  1. Please justify the sample size in the study.

Authors: Thanks. We have corrected that. The sample size was calculated using 80 % statistical power and 95 % confidence interval. The mean and standard deviation for the APAonset (s) was estimated in a pilot study performed in the first 7 subjects in two sessions. The mean difference between sessions was achieved as 0.011905 ± 0.011649s. A required sample size of 10 individuals was calculated and authors decided to test the double of this minimum. This information was included in “Methods”, in the new version.

  1. Line 171: reference 17 is inappropriate here an should be deleted as it only cites reference 13.

Authors: Thanks. We have corrected that.

DISCUSSION

  1. Lines 247-249: this is just repetition of the results

Authors: Thanks. We have corrected that.

  1. Please provide information on how your results will impact research and/or clinical practice.
  2. Please discuss the generalizability of the results to the athlete population.

Authors: Thanks. We have rewritten the discussion, providing information on how your results will impact research and/or clinical practice and on the generalizability of the results to the athlete population. We hope it is better now.

CONCLUSIONS

  1. Line 262: given your study design/population, this is an overestimation – please delete/rephrase.

Authors: Thanks. We have rewritten the conclusion and hope it is better now.

Round 2

Reviewer 1 Report

The authors have responded to all my suggestions. I congratulate the authors for their work. 

Reviewer 2 Report

The authors have addressed my comments satisfactorily in the revised manuscript. I now recommend accpectance of the paper.

Reviewer 3 Report

Authors fully addressed my comments.